# Crafting Interpretable Embeddings for Language Neuroscience by Asking LLMs Questions

**Vinamra Benara***
UC Berkeley

**Chandan Singh***
Microsoft Research

**John X. Morris**
Cornell University

**Richard J. Antonello**
UT Austin

**Ion Stoica**
UC Berkeley

**Alexander G. Huth**
UT Austin

**Jianfeng Gao**
Microsoft Research

*Equal contribution

## Abstract

Large language models (LLMs) have rapidly improved text embeddings for a growing array of natural-language processing tasks. However, their opaqueness and proliferation into scientific domains such as neuroscience have created a growing need for interpretability. Here, we ask whether we can obtain interpretable embeddings through LLM prompting. We introduce question-answering embeddings (QA-Emb), embeddings where each feature represents an answer to a yes/no question asked to an LLM. Training QA-Emb reduces to selecting a set of underlying questions rather than learning model weights.

We use QA-Emb to flexibly generate interpretable models for predicting fMRI voxel responses to language stimuli. QA-Emb significantly outperforms an established interpretable baseline, and does so while requiring very few questions. This paves the way towards building flexible feature spaces that can concretize and evaluate our understanding of semantic brain representations. We additionally find that QA-Emb can be effectively approximated with an efficient model, and we explore broader applications in simple NLP tasks.[1]

## 1 Introduction

Text embeddings are critical to many applications, including information retrieval, semantic clustering, retrieval-augmented generation, and language neuroscience. Traditionally, text embeddings leveraged interpretable representations such as bag-of-words or BM-25 [1]. Modern methods often replace these embeddings with representations from large language models (LLMs), which may better capture nuanced contexts and interactions [2–7]. However, these embeddings are essentially black-box representations, making it difficult to understand the predictive models built on top of them (as well as why they judge different texts to be similar in a retrieval context). This opaqueness is detrimental in scientific fields, such as neuroscience [8] or social science [9], where trustworthy interpretation itself is the end goal. Moreover, this opaqueness has debilitated the use of LLM embeddings (for prediction or retrieval) in high-stakes applications such as medicine [10], and raised issues related to regulatory pressure, safety, and alignment [11–14].

To ameliorate these issues, we introduce question-answering embeddings (QA-Emb), a method that builds an interpretable embedding by repeatedly querying a pre-trained autoregressive LLM with a set of questions that are selected for a problem (Fig. 1). Each element of the embedding represents

---

[1]All code for QA-Emb is made available on Github at  github.com/csinva/interpetable-embeddings.

38th Conference on Neural Information Processing Systems (NeurIPS 2024).

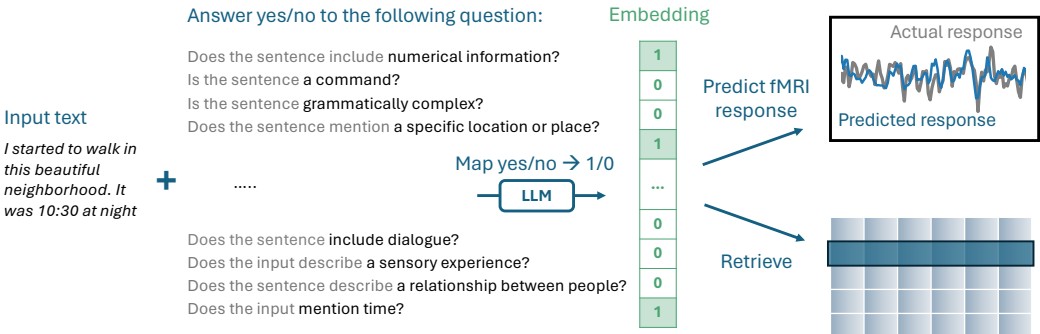

Figure 1: QA-Emb produces an embedding for an input text by prompting an LLM with a series of yes/no questions. This embedding can then be used in downstream tasks such as fMRI response prediction or information retrieval.

the answer to a different question asked to an LLM, making the embedding human-inspectable. For example, the first element may be the answer to the question *Does the input mention time?* and the output would map *yes/no* to 1/0. Training QA-Emb requires only black-box access to the LLM (it does not require access to the LLM internals) and modifies only natural-language prompts, rather than LLM parameters. The learning problem is similar to the optimization faced in natural-language autoprompting [15, 16] or single-neuron explanation [17, 18], but seeks a set of questions rather than an individual prompt.

We focus on a single neuroscience problem in close collaboration with neuroscientists. Grounding in a neuroscience context allows us to avoid common pitfalls in evaluating interpretation methods [19, 20] that seek to test "interpretability" generally. Additionally, this focus allows to more realistically integrate domain knowledge to select and evaluate the questions needed for QA-Emb, one of its core strengths. Nevertheless, QA-Emb may be generally applicable in other domains where it is important to meaningfully interpret text embeddings.

In our neuroscience setting, we build QA-Emb representations from natural-language questions that can predict human brain responses measured by fMRI to natural-language stimuli. This allows for converting informal verbal hypotheses about the semantic selectivity of the brain into quantitative models, a pressing challenge in fields such as psychology [21]. We find that predictive models built on top of QA-Embs are quite accurate, providing a 26% improvement over an established interpretable baseline [22] and even slightly outperforming a black-box BERT baseline [23]. Additionally, QA-Emb yields concise embeddings, outperforming the interpretable baseline (that consists of 985 features) with only 29 questions.

We investigate two major limitations of QA-Emb in Sec. 5. First, with regards to computational efficiency, we find that we can drastically reduce the computational cost of QA-Emb by distilling it into a model that computes the answers to all selected questions in a single feedforward pass by using many classification heads. Second, we evaluate the accuracy of modern LLMs at reliably answering diverse yes/no questions. Finally, Sec. 6 explores broader applications for QA-Emb in a simple information retrieval setting and text-clustering setting.

## 2   Methods

QA-Emb is an intuitive method to generate text embeddings from a pre-trained autoregressive LLM (Fig. 1). Given a text input, QA-Emb builds an interpretable embedding by querying the LLM with a set of questions about the input. Each element of the embedding represents the answer to a different question asked to an LLM. This procedure allows QA-Emb to capture nuanced and relevant details in the input while staying interpretable.

**Learning a set of yes/no questions**   QA-Emb requires specifying a set of yes/no questions $Q \in \mathcal{Q}_{\text{yes/no}}$ that yield a binary embedding $v_Q(x) \in \{0, 1\}^d$ for an input string $x$. The questions are chosen to yield embeddings that are suitable for a downstream task. In our fMRI prediction task, we optimize for supervised linear regression: given a list of $n$ input strings $X$ and a multi-dimensional continuous output $Y \in \mathbb{R}^{nxd}$, we seek embeddings that allow for learning effective ridge regression models:

$$Q = \underset{Q \in \mathcal{Q}_{\text{yes/no}}}{\text{argmin}} \left[ \min_{\theta \in \mathbb{R}^d} \sum_i^n ||Y^{(i)} - \theta^T v_Q(X^{(i)})|| + \lambda ||\theta||_2 \right], \tag{1}$$

where $\theta$ is a learned coefficient vector for predicting the fMRI responses and $\lambda$ is the ridge regularization parameter.

Directly optimizing over the space of yes/no questions is difficult, as it requires searching over a discrete space with a constraint set $\mathcal{Q}_{\text{yes/no}}$ that is hard to specify. Instead, we heuristically optimize the set of questions $Q$, by prompting a highly capable LLM (e.g. GPT-4 [24]) to generate questions relevant to our task, e.g. *Generate a bulleted list of questions with yes/no answers that is relevant for {{task description}}*. Customizing the task description helps yield relevant questions. The prompt can flexibly specify more prior information when available. For example, it can include examples from the input dataset to help the LLM identify data-relevant questions. Taking this a step further, questions can be generated sequentially (similar to gradient boosting) by having the LLM summarize input examples that incur high prediction error to generate new questions focused on those examples. While we focus on optimizing embeddings for fMRI ridge regression in Eq. (1), different downstream tasks may require different inner optimization procedures, e.g. maximizing the similarity of relevant documents for retrieval.

**Post-hoc pruning of $Q$.**   The set of learned questions $Q$ can be easily pruned to be made compact and useful in different settings. For example, in our fMRI regression setting, a feature-selection procedure such as Elastic net [25] can be used to remove redundant/uninformative questions from the specified set of questions $Q$. Alternatively, an LLM can be used to directly adapt $Q$ to yield task-specific embeddings. Since the questions are all in natural language, they can be listed in a prompt, and an LLM can be asked to filter the task-relevant ones, e.g. *Here is a list of questions:{{question list}} List the subset of these questions that are relevant for {{task description}}*.

**Limitations: computational cost and LLM inaccuracies.**   While effective, the QA-Emb pipeline described here has two major limitations. First, QA-Emb is computationally intensive, requiring $d$ LLM calls to compute an embedding. This is often prohibitively expensive, but may be worthwhile in high-value applications (such as our fMRI setting) and will likely become more tenable as LLM inference costs continue to rapidly decrease. We find that we can dramatically reduce this cost by distilling the QA-Emb model into a single LLM model with many classification heads in Sec. 5.1. Otherwise, LLM inference costs are partially mitigated by the ability to reuse the KV-cache for each question and the need to only generate a single token for each question. While computing embeddings with QA-Emb is expensive, *searching* embeddings is made faster by the fact that the resulting embeddings are binary and often relatively compact.

Second, QA-Emb requires that the pre-trained LLM can faithfully answer the given yes-no questions. If an LLM is unable to accurately answer the questions, it hurts explanation's faithfulness. Thus, QA-Emb requires the use of fairly strong LLMs and the set of chosen questions should be accurately answered by these LLMs (Sec. 5.2 provides analysis on the question-answering accuracy of different LLMs).

**Hyperparameter settings**   For answering questions, we average the answers from Mistral-7B [26] (`mistralai/Mistral-7B-Instruct-v0.2`) and LLaMA-3 8B [27] (`meta-llama/Meta-Llama-3-8B-Instruct`) with two prompts. All perform similarly and averaging their answers yields a small performance improvement (Table A2). For generating questions, we prompt GPT-4 [24] (`gpt-4-0125-preview`). Experiments were run using 64 AMD MI210 GPUs, each with 64 gigabytes of memory, and reproducing all experiments in the paper requires approximately 4 days (initial explorations required roughly 5 times this amount of compute). All prompts used and generated questions are given in the appendix or on Github.

## 3 Related work

**Text embeddings**   Text embeddings models, which produce vector representations of document inputs, have been foundational to NLP. Recently, transformer-based models have been trained to yield embeddings in a variety of ways [2–7], including producing embeddings that are sparse [28] or have variable lengths [29]. Recent works have also leveraged autoregressive LLMs to build embeddings, e.g. by repeating embeddings [30], generating synthetic data [6, 31], or using the last-token distribution of an autoregressive LLM as an embedding [32]. Similar to QA-Emb, various works have used LLM answers to multiple prompts for different purposes, e.g. text classification [33–35], learning style embeddings [36], or data exploration [37].

**Interpreting representations**   A few works have focused on building intrinsically interpretable text representations, e.g. word or ngram-based embeddings such as word2vec [38], Glove [39], and LLM word embeddings. Although their dimensions are not natively interpretable, for some tasks, such as classification, they can be projected into a space that is interpretable [40], i.e. a word-level representation. Note that it is difficult to learn a sparse interpretable model from these dense embeddings, as standard techniques (e.g. Elastic net) cannot be directly applied.

When instead using black-box representations, there are many post-hoc methods to interpret embeddings, e.g. probing [41, 42], categorizing elements into categories [43–46], categorizing directions in representation space [47–50], or connecting multimodal embeddings with text embeddings/text concepts [51–55]. For a single pair of text embeddings, prediction-level methods can be applied to approximately explain why the two embeddings are similar [56, 57].

**Natural language representations in fMRI**   Using LLM representations to help predict brain responses to natural language has recently become popular among neuroscientists studying language processing [58–63] (see [64, 65] for reviews). This paradigm of using "encoding models" [66] to better understand how the brain processes language has been applied to help understand the cortical organization of language timescales [67, 68], examine the relationship between visual and semantic information in the brain [69], and explore to what extent syntax, semantics, or discourse drives brain activity [22, 70–77, 18]. The approach here extends these works to build an increasingly flexible, interpretable feature space for modeling fMRI responses to text data.

## 4 Main results: fMRI interpretation

A central challenge in neuroscience is understanding how and where semantic concepts are represented in the brain. To meet this challenge, we extend the line of study that fits models to predict the response of different brain voxels (i.e. small regions in the brain) to natural language stimuli. Using QA-Emb, we seek to bridge models that are interpretable [1, 22] with more recent LLM models that are accurate but opaque [58–60].

### 4.1   fMRI experimental setup

**Dataset**   We analyze data from two recent studies [78, 79] (released under the MIT license), which contain fMRI responses for 3 human subjects listening to 20+ hours of narrative stories from podcasts. We extract text embeddings from the story that each subject hears and fit a ridge regression to predict the fMRI responses (Eq. (1)). Each subject listens to either 79 or 82 stories (consisting of 27,449 time points) and 2 test stories (639 time points); Each subject's fMRI data consists of approximately 100,000 voxels; we preprocess it by running principal component analysis (PCA) and extracting the coefficients of the top 100 components.

**Regression modeling**   We fit ridge regression models to predict these 100 coefficients and evaluate the models in the original voxel space (by applying the inverse PCA mapping and measuring the correlation between the response and prediction for each voxel). We deal with temporal sampling following [22, 60]; an embedding is produced at the timepoint for each word in the input story and these embeddings are interpolated using Lanczos resampling. Embeddings at each timepoint are produced from the ngram consisting of the 10 words preceding the current timepoint. We select the best-performing hyperparameters via cross-validation on 5 time-stratified bootstrap samples of the training set. We select the best ridge parameters from 12 logarithmically spaced values between 10

and 10,000. To model temporal delays in the fMRI signal, we also select between adding 4, 8, or 12 time-lagged duplicates of the stimulus features.

**Generating QA-Emb questions**   To generate the questions underlying QA-Emb, we prompt GPT-4 with 6 prompts that aim to elicit knowledge useful for predicting fMRI responses (precise prompts in Appendix A.3). This includes directly asking the LLM to use its knowledge of neuroscience, to brainstorm semantic properties of narrative sentences, to summarize examples from the input data, and to generate questions similar to single-voxel explanations found in a prior work [18]. This process yields 674 questions (Fig. 1 and Table A1 show examples, see all questions on Github). We perform feature selection by running multi-task Elastic net with 20 logarithmically spaced regularization parameters ranging from $10^{-3}$ to 1 and then fit a Ridge regression to the selected features.[2] See extended details on the fMRI experimental setup in Appendix A.1 and all prompts in Appendix A.3.

**Baselines**   We compare QA-Emb to Eng1000, an interpretable baseline developed in the neuroscience literature specifically for the task of predicting fMRI responses from narrative stories [22]. Each element in an Eng1000 embedding corresponds to a cooccurence statistic with a different word, allowing full interpretation of the underlying representation in terms of related words. We additionally compare to embeddings from BERT [23] (`bert-base-uncased`) and LLaMA models [81, 27]. For each subject, we sweep over 5 layers from LLaMA-2 7B (`meta-llama/Llama-2-7b-hf`, layers 6, 12, 18, 24, 30), LLaMA-2 70B (`meta-llama/Llama-2-70b-hf`, layers 12, 24, 36, 48, 60), and LLaMA-3 8B (`meta-llama/Meta-Llama-3-8B`, layers 6, 12, 18, 24, 30), then report the test performance for the model that yields the best cross-validated accuracy (see breakdown in Table A3).

## 4.2   fMRI predictive performance

We find that QA-Emb predicts fMRI responses fairly well across subjects (Fig. 2A), achieving an average test correlation of 0.116. QA-Emb significantly outperforms the interpretable baseline Eng1000 (26% average improvement). Comparing to the two transformer-based baselines (which do not yield straightforward interpretations), we find that QA-Emb slightly outperforms BERT (5% improvement) and worse than the best cross-validated LLaMA-based model (7% decrease). Trends are consistent across all 3 subjects.

To yield a compact and interpretable model, Fig. 2B further investigates the compressibility of the two interpretable methods (through Elastic net regularization). Compared to Eng1000, QA-Emb improves performance very quickly as a function of the number of features included, even outperforming the final Eng1000 performance with only 29 questions (mean test correlation 0.122 versus 0.118). Table A1 shows the 29 selected questions, which constitute a human-readable description of the entire model.

Fig. 2C-D further break down the predictive performance across different brain regions for a particular subject (S03). The regions that are well-predicted by QA-Emb (Fig. 2C) align with language-specific areas that are seen in the literature [59, 82]. They do not show any major diversions from transformer-based encoding models (Fig. 2D), with the distribution of differences being inconsistent across subjects (see Fig. A1).

## 4.3   Interpreting the fitted representation from QA-Emb

The QA-Emb representation enables not only identifying which questions are important for fMRI prediction, but also mapping their selectivity across the cortex. We analyze the QA-Emb model which uses 29 questions and visualize the learned regression weights for different questions. Fig. 3 shows example flatmaps of the regression coefficients for 3 of the questions across the 2 best-predicted subjects (S02 and S03). Learned feature weights for the example questions capture known selectivity and are highly consistent across subjects. In particular, the weights for the question "*Does the sentence involve a description of a physical environment or setting*?" captures classical place areas including occipital place area [83] and retrosplenial complex [84], as well as intraparietal sulcus [85]. The weights for the question "*Is the sentence grammatically complex*?" bear striking similarity to the language network [82, 86], which is itself localized from a contrast between sentences and nonwords. Other questions, such as "*Does the sentence describe a physical action*?", which has strong right

---

[2]We run Elastic net using the MultiTaskElasticNet class from scikit-learn [80].

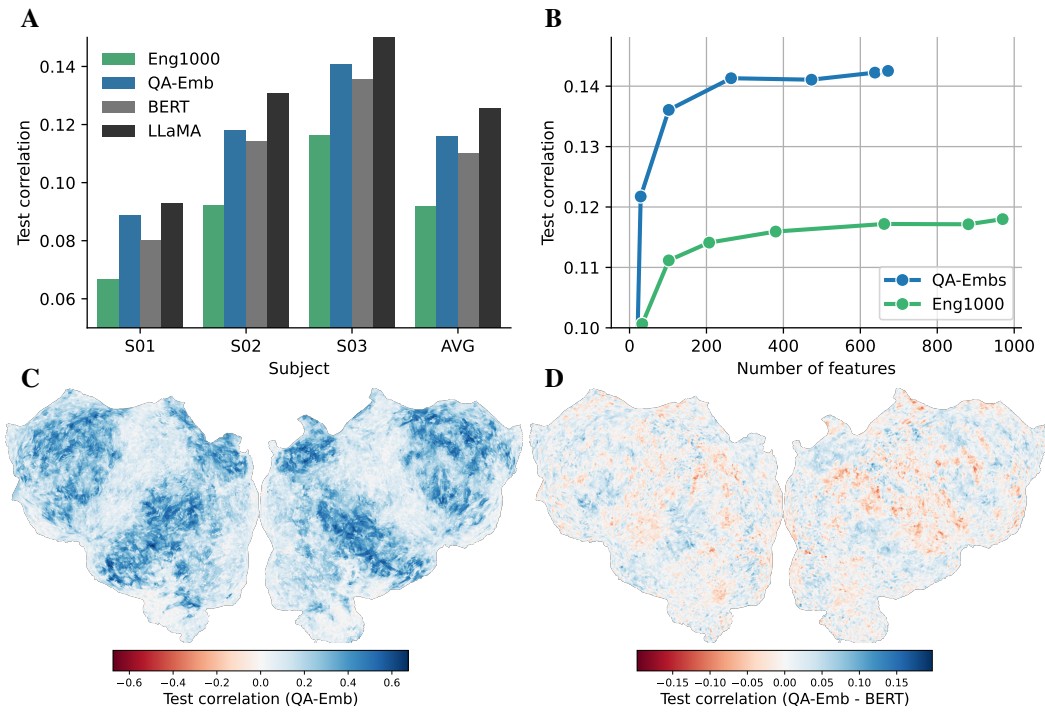

Figure 2: Predictive performance for QA-Emb compared to baselines. (A) Test correlation for QA-Emb outperforms the interpretable Eng1000 baseline, is on par with the black-box BERT baseline, and is worse than the best-performing LLaMA model. (B) Test correlation for method quickly grows as a function of the number of included questions. (C) Test correlation per voxel for QA-Emb. (D) Difference in the test correlation per voxel for subject between QA-Emb and BERT. Error bars for (A) and (B) (standard error of the mean) are within the points (all are below 0.001). (B), (C), and (D) show results for subject S03.

Table 1: Mean test correlation when comparing QA-Emb computed via many LLM calls to QA-Emb computed via a single distilled model. Distillation does not significantly degrade performance. All standard errors of the mean are below $10^{-3}$.

|  | QA-Emb | QA-Emb (distill, binary) | QA-Emb (distill, probabilistic) | Eng1000 |
|---|---|---|---|---|
| UTS01 | 0.081 | 0.083 | 0.080 | 0.077 |
| UTS02 | 0.124 | 0.118 | 0.118 | 0.096 |
| UTS03 | 0.136 | 0.132 | 0.142 | 0.117 |
| **AVG** | **0.114** | **0.111** | **0.113** | **0.097** |

laterality, do not have a strong basis in prior literature. These questions point to potentially new insights into poorly understood cortical regions.

## 5 Evaluating the limitations of QA-Emb

### 5.1 Improving computational efficiency via model distillation

To reduce the computational cost of running inference with QA-Emb, we explore distilling the many LLM calls needed to compute QA-Emb into a single model with many classification heads. Specifically, we finetune a RoBERTa model [87] (`roberta-base`) with 674 classification heads to predict all answers required for QA-Emb in a single feedforward pass. We finetune the model on answers from LLaMA-3 8B with a few-shot prompt for 80% of the 10-grams in the 82 fMRI training stories (123,203 examples), use the remaining 20% as a validation set for early stopping (30,801

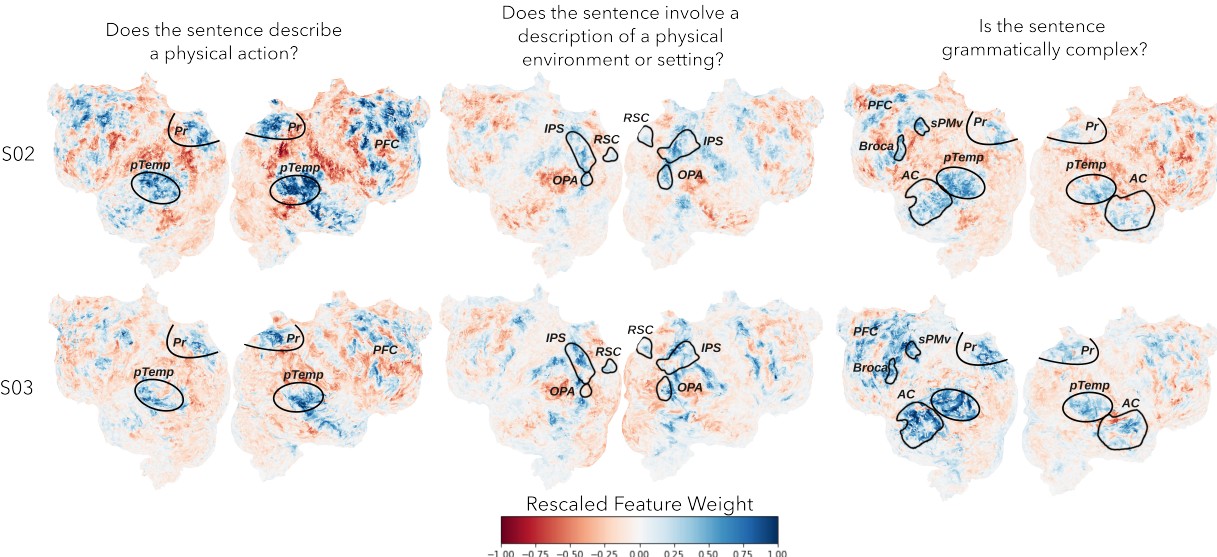

Figure 3: Learned feature weights for 3 example questions capture known selectivity and are consistent across subjects. All feature weights are jointly rescaled to the range (-1, 1) for visualization. Abbreviations: Pr = precuneus, pTemp = posterior temporal cortex, PFC = prefrontal cortex, IPS = intraparietal sulcus, RSC = retrosplenial complex, OPA = occipital place area, PPA = parahippocampal place area, Broca = Broca's area, sPMv = superior premotor ventral speech area, AC = auditory cortex.

examples), and evaluate on all 10-grams in the 2 testing stories (4,594 examples). We finetune using AdamW [88] with a learning rate of $5 \cdot 10^{-5}$.

When evaluated on the fMRI prediction task, the distilled model (*QA-Emb (distill, binary)* in Table 1) yields a performance only slightly below the original model. If we relax the restriction that the finetuned model yields binary embeddings and instead use the predicted probability for *yes*, the performance rises slightly to nearly match the original model (0.113 instead of 0.114 average test correlation) and maintains a significant improvement over the Eng1000 baseline. Note that the distilled model achieves an 88.5% match for yes/no answers on 10-grams for the test set. Nevertheless, the fMRI prediction for any given timepoint is computed from many questions and ngrams, mitigating the effect of individual errors in answering a question.

## 5.2 Evaluating question-answering faithfulness

We evaluate the faithfulness of our question-answering models on a recent diverse collection of 54 binary classification datasets [89, 90] (see data details in Table A4). These datasets are difficult, as they are intended to encompass a wider-ranging and more realistic list of questions than traditional NLP datasets.

Fig. 4 shows the classification accuracy for the 3 LLMs used previously along with GPT-3.5 (gpt-3.5-turbo-0125). On average, each of the LLMs answers these questions with fairly high accuracy, with GPT-4 slightly outperforming the other models. However, we observe poor performance on some tasks, which we attribute to the task difficulty and the lack of task-specific prompt engineering. For example, the dataset yielding the lowest accuracy asks the question *Is the input about math research?*. While this may seem like a fairly simple question for an LLM to answer, the examples in the negative class consist of texts from other quantitative fields (e.g. chemistry) that usually contain numbers, math notation, and statistical analysis. Thus the LLMs answer *yes* to most examples and achieve accuracy near chance (50%). Note that these tasks are more difficult than the relatively simple questions we answer in the fMRI experiments, especially since the fMRI input lengths are each 10 words, whereas the input lengths for these datasets are over 50 words on average (with some inputs spanning over 1,000 words).

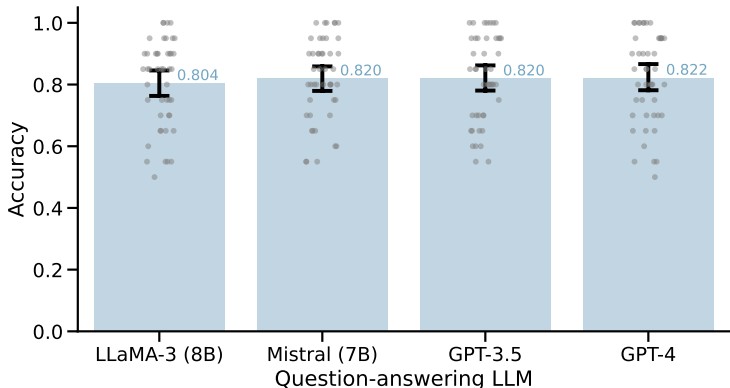

Figure 4: Performance of question-answering for underlying LLMs on the D3 collection of binary classification datasets. Each point shows an individual dataset and error bars show the 95% confidence interval.

Table 2: Information retrieval results for different interpretable embedding models. QA-Emb in combination with BM-25 achieves a slight improvement over the interpretable baselines. QA-Emb additionally yields reasonably strong performance compared to its embedding size. [†]Note that QA-Emb embeddings are binary, so the raw number of dimensions overrepresents the embedding's size relative to other methods. Error bars show standard error of the mean.

|  | Mean reciprocal rank | Recall@1 | Recall@5 | Size |
|---|---|---|---|---|
| Bag of words | 0.37±.01 | 0.28±.02 | 0.42±.02 | 27,677 |
| Bag of bigrams | 0.39±.01 | 0.30±.02 | 0.44±.02 | 197,924 |
| Bag of trigrams | 0.39±.02 | 0.30±.02 | 0.44±.02 | 444,403 |
| QA-Emb | 0.45±.01 | 0.34±.01 | 0.50±.01 | [†]**2,000** |
| BM-25 | 0.77±.01 | 0.69±.01 | 0.82±.01 | 27,677 |
| **BM-25 + QA-Emb** | **0.80±.01** | **0.71±.01** | **0.84±.01** | 29,677 |

## 6   Secondary results: evaluating QA-Emb in simple NLP tasks

### 6.1   Benchmarking QA-Emb for information retrieval

In this section, we investigate applying QA-Emb to a simplified information retrieval task. We take a random subset of 4,000 queries from the MSMarco dataset ([91], Creative Commons License) and their corresponding groundtruth documents, resulting in 5,210 documents. We use 25% of the queries to build a training set and keep the remaining 75% for testing. For evaluation, we calculate the cosine similarity match between the embeddings for each query and its groundtruth documents using mean reciprocal rank and recall.

To compute QA-Emb, we first generate 2,000 questions through prompting GPT-4 based on its knowledge of queries in information retrieval (see prompts in the Github). We use a regex to slightly rewrite the resulting questions for queries to apply to documents (e.g. *Is this query related to a specific timeframe?* → *Is this text related to a specific timeframe?*). We then answer the questions both for each query and for each corpus document, again using LLaMA-3 8B. Rather than fitting a ridge regression as in Eq. (1), we use the training set to learn a scalar for each question that multiplies its binary output to change both its sign and magnitude in the embedding (optimization details in Appendix A.4).

Table 2 shows the information retrieval results. Combining BM-25 with QA-Emb achieves a small but significant improvement over the interpretable baselines. QA-Emb on its own achieves modest performance, slightly improving slightly over a bag-of-words representation, but significantly underperforming BM-25. Nevertheless, its size is considerably smaller than the other interpretable baselines making it quicker to interpret and to use for retrieval.

Table 3: Clustering scores before and after zero-shot adaptation (higher is better). Errors give standard error of the mean.

| | Rotten tomatoes | AG News | Emotion | Financial phrasebank | AVG | Embedding size (AVG) |
|---|---|---|---|---|---|---|
| Original | 0.126±0.011 | 0.124±0.007 | 0.046±0.007 | 0.084±0.008 | 0.095 | 100 |
| **Adapted** | **0.248±0.016** | **0.166±0.012** | **0.057±0.010** | **0.292±0.017** | **0.191** | **25.75±0.95** |

## 6.2 Zero-shot adaptation in text clustering

We now investigate QA-Emb in a simplified text clustering setting. To do so, we study 4 text-classification datasets: Financial phrasebank ([92], creative commons license), Emotion [93] (CC BY-SA 4.0 license), AGNews [94], and Rotten tomatoes [95]. For each dataset, we treat each class as a cluster and evaluate the *clustering score*, defined as the difference between the average inter-class embedding distance and the average intra-class embedding distance (embedding distance is measured via Euclidean distance). A larger clustering score suggests that embeddings are well-clustered within each class.

In our experiment, we build a 100-dimensional embedding by prompting GPT-4 to generate 25 yes/no questions related to the semantic content of each dataset (e.g. for Rotten tomatoes, *Generate 25 yes/no questions related to movie reviews*). We then concatenate the answers for all 100 questions to form our embedding. These general embeddings do not yield particularly strong clustering scores (Table 3 top), as the questions are diverse and not particularly selective for each dataset.

However, simply through prompting, we can adapt these general embeddings to each individual dataset. We call GPT-4 with a prompt that includes the full list of questions and ask it to select a subset of questions that are relevant to each task. The result embeddings (Table 3 bottom) yield higher clustering scores, suggesting that QA-Emb can be adapted to each task in a zero-shot manner (in this simplified setting). Moreover, the resulting task-specific embeddings are now considerably smaller.

## 7 Discussion

We find that QA-Emb can effectively produce interpretable and high-performing text embeddings. While we focus on a language fMRI setting, QA-Emb may be able to help flexibly build an interpretable text feature space in a variety of domains, such as social science [9], medicine [10], or economics [96], where meaningful properties of text can help discover something about an underlying phenomenon or build trust in high-stakes settings. Alternatively, it could be used in mechanistic interpretability, to help improve post-hoc explanations of learned LLM representations.

As LLMs improve in both efficiency and capability, QA-Emb can be incorporated into a variety of common NLP applications as well, such as RAG or information retrieval. For example, in RAG systems such as RAPTOR [97] or Graph-RAG [98], explanations may help an LLM not only retrieve relevant texts, but also specify why they are relevant and how they may be helpful.

Learning text questions rather than model weights is a challenging research area, furthering work in automatic prompt engineering [15, 16]. Our approach takes a heuristic first step at solving this problem, but future work could explore more directly optimizing the set of learned questions $Q$ in Eq. (1) via improved discrete optimization approaches and constraints. One possible approach may involve having LLMs themselves identify the errors the current model is making and improving based on these errors, similar to general trends in LLM self-improvement and autoprompting [99–102]. Another approach may involve improving the explanation capabilities of LLMs to help extract more questions more faithfully from data [103, 104].

**Broader Impacts** QA-Emb seeks to advance the field of LLM interpretation, a crucial step toward addressing the challenges posed by these often opaque models. Although LLMs have gained widespread use, their lack of transparency can lead to significant harm, underscoring the importance of interpretable AI. There are many potential positive societal consequences of this form of interpretability, e.g., facilitating a better understanding of scientific data and models, along with a better understanding of LLMs and how to use them safely. Nevertheless, as is the case with most ML research, the interpretations could be used to interpret and potentially improve an LLM or dataset

that is being used for nefarious purposes. Moreover, QA-Emb requires substantial computational resources, contributing to increased concerns over sustainability.

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

# A Appendix

## A.1 fMRI question details

Table A1: Questions list for model with 29 questions. Importance denotes the average absolute coefficient for each question (normalized by the importance of the top question).

| Question | Importance |
|---|---|
| Is the sentence expressing skepticism or disbelief towards something or someone? | 1.000 |
| Does the sentence include dialogue? | 0.983 |
| Does the sentence describe a relationship between people? | 0.924 |
| Does the sentence involve the mention of a specific object or item? | 0.900 |
| Does the sentence include technical or specialized terminology? | 0.882 |
| Does the sentence contain a proper noun? | 0.861 |
| Does the input involve planning or organizing? | 0.861 |
| Does the sentence include numerical information? | 0.850 |
| Is time mentioned in the input? | 0.844 |
| Is the sentence grammatically complex? | 0.815 |
| Does the sentence include dialogue or thoughts directed towards another character? | 0.811 |
| Does the sentence describe a physical action? | 0.809 |
| Does the sentence include a conditional clause? | 0.782 |
| Does the sentence describe a visual experience or scene? | 0.771 |
| Does the input include a philosophical or reflective thought? | 0.759 |
| Is the sentence conveying the narrator's physical movement or action in detail? | 0.749 |
| Does the sentence describe a physical sensation? | 0.744 |
| Does the sentence involve a discussion about personal or social values? | 0.739 |
| Does the sentence reference a specific time or date? | 0.719 |
| Does the sentence express a philosophical or existential query or observation? | 0.705 |
| Does the sentence involve a description of physical environment or setting? | 0.693 |
| Does the input describe a sensory experience? | 0.688 |
| Does the sentence involve planning or decision-making? | 0.684 |
| Is the sentence a command? | 0.682 |
| Does the sentence describe a specific sensation or feeling? | 0.672 |
| Does the sentence contain a cultural reference? | 0.667 |
| Does the input include dialogue between characters? | 0.594 |
| Does the sentence mention a specific location or place? | 0.547 |
| Does the sentence reference a specific location or place? | 0.545 |

## A.2 fMRI prediction results extended

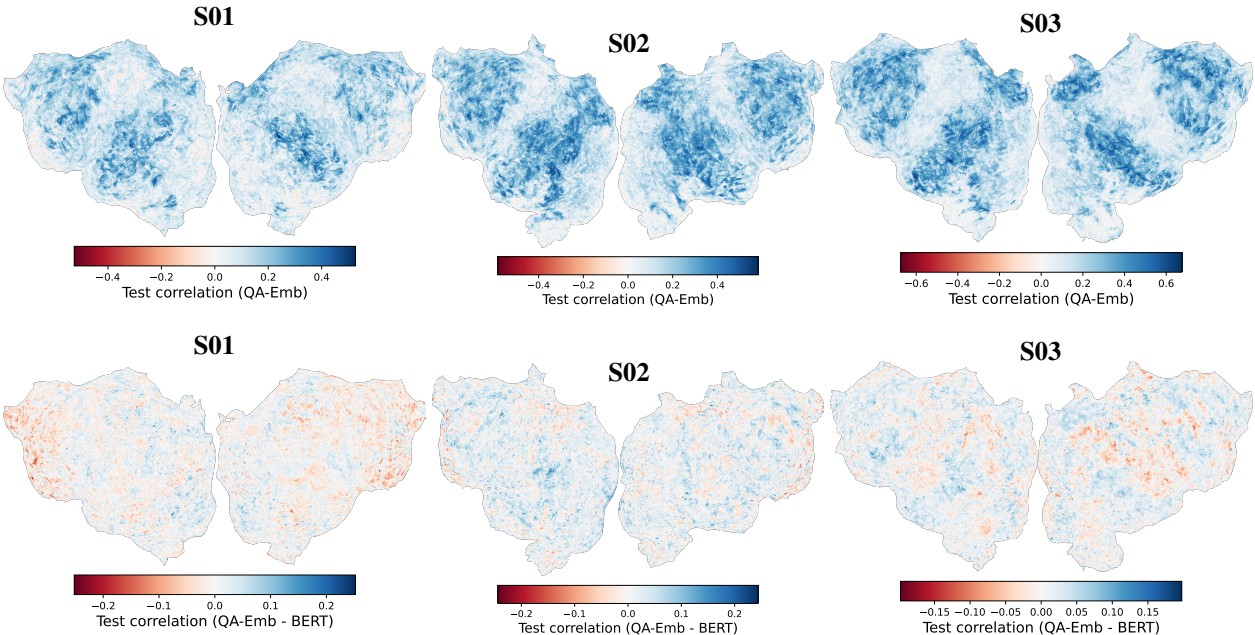

Figure A1: Predictive performance for QA-Emb (top row) and the difference between QA-Emb and BERT (bottom row).

Table A2: Mean test correlation for QA-Emb with different settings: varying the underlying prompts to source questions and the LLM used to answer the questions (fixing the number of time-lagged delays to 8). Ensemble generally provides a small boost over other models and Mistral slightly underperforms LLaMA-3 (8B).

| Subject | Questions | Ensemble | LLaMA-3 (8B) | LLaMA-3 (8B)-fewshot | Mistral (7B) |
|---|---|---|---|---|---|
| S01 | Prompts 1-3 (376 questions) | 0.081 | 0.078 | 0.078 | 0.076 |
| | Prompts 1-5 (518 questions) | 0.089 | 0.085 | 0.085 | 0.082 |
| | Prompts 1-6 (674 questions) | 0.084 | 0.081 | 0.085 | 0.076 |
| S02 | Prompts 1-3 (376 questions) | 0.120 | 0.112 | 0.119 | 0.112 |
| | Prompts 1-5 (518 questions) | 0.118 | 0.120 | 0.121 | 0.114 |
| | Prompts 1-6 (674 questions) | 0.124 | 0.119 | 0.121 | 0.108 |
| S03 | Prompts 1-3 (376 questions) | 0.132 | 0.131 | 0.127 | 0.126 |
| | Prompts 1-5 (518 questions) | 0.137 | 0.136 | 0.135 | 0.129 |
| | Prompts 1-6 (674 questions) | 0.141 | 0.136 | 0.136 | 0.132 |
| AVG | Prompts 1-3 (376 questions) | 0.111 | 0.107 | 0.108 | 0.104 |
| | Prompts 1-5 (518 questions) | 0.115 | 0.114 | 0.114 | 0.108 |
| | Prompts 1-6 (674 questions) | 0.116 | 0.112 | 0.114 | 0.105 |

Table A3: Mean test correlation for different baseline models as a function of hyperparameters (number of time-lagged delays and layer for extracting embeddings)

| Subject | S01 | | | S02 | | | S03 | | | AVG | | |
|---|---|---|---|---|---|---|---|---|---|---|---|---|
| Delays | 4 | 8 | 12 | 4 | 8 | 12 | 4 | 8 | 12 | 4 | 8 | 12 |
| BERT | 0.084 | 0.080 | 0.075 | 0.114 | 0.108 | 0.107 | 0.136 | 0.139 | 0.136 | 0.111 | 0.109 | 0.106 |
| Eng1000 | 0.079 | 0.067 | 0.077 | 0.096 | 0.092 | 0.082 | 0.110 | 0.117 | 0.116 | 0.095 | 0.092 | 0.092 |
| LLaMA-2 (70B) (lay 12) | 0.055 | 0.055 | 0.054 | 0.101 | 0.095 | 0.085 | 0.143 | 0.144 | 0.130 | 0.100 | 0.098 | 0.089 |
| LLaMA-2 (70B) (lay 24) | 0.075 | 0.059 | 0.049 | 0.097 | 0.104 | 0.092 | 0.149 | 0.153 | 0.152 | 0.107 | 0.105 | 0.098 |
| LLaMA-2 (70B) (lay 36) | 0.058 | 0.068 | 0.057 | 0.131 | 0.101 | 0.084 | 0.153 | 0.156 | 0.152 | 0.114 | 0.108 | 0.098 |
| LLaMA-2 (70B) (lay 48) | 0.093 | 0.060 | 0.052 | 0.114 | 0.094 | 0.091 | 0.148 | 0.151 | 0.149 | 0.118 | 0.102 | 0.098 |
| LLaMA-2 (70B) (lay 60) | 0.095 | 0.048 | 0.050 | 0.119 | 0.089 | 0.088 | 0.148 | 0.152 | 0.150 | 0.121 | 0.097 | 0.096 |
| LLaMA-2 (7B) (lay 06) | 0.074 | 0.067 | 0.039 | 0.120 | 0.088 | 0.084 | 0.138 | 0.144 | 0.133 | 0.111 | 0.100 | 0.085 |
| LLaMA-2 (7B) (lay 12) | 0.097 | 0.058 | 0.053 | 0.116 | 0.111 | 0.087 | 0.150 | 0.155 | 0.152 | 0.121 | 0.108 | 0.097 |
| LLaMA-2 (7B) (lay 18) | 0.079 | 0.076 | 0.042 | 0.123 | 0.103 | 0.090 | 0.143 | 0.153 | 0.150 | 0.115 | 0.111 | 0.094 |
| LLaMA-2 (7B) (lay 24) | 0.088 | 0.057 | 0.068 | 0.129 | 0.100 | 0.106 | 0.144 | 0.148 | 0.149 | 0.120 | 0.102 | 0.108 |
| LLaMA-2 (7B) (lay 30) | 0.057 | 0.045 | 0.045 | 0.130 | 0.098 | 0.099 | 0.139 | 0.149 | 0.148 | 0.109 | 0.097 | 0.097 |
| LLaMA-3 (8B) (lay 06) | 0.071 | 0.066 | 0.054 | 0.122 | 0.119 | 0.095 | 0.144 | 0.147 | 0.148 | 0.112 | 0.111 | 0.099 |
| LLaMA-3 (8B) (lay 12) | 0.089 | 0.073 | 0.050 | 0.110 | 0.099 | 0.095 | 0.146 | 0.151 | 0.153 | 0.115 | 0.108 | 0.099 |
| LLaMA-3 (8B) (lay 18) | 0.073 | 0.052 | 0.052 | 0.125 | 0.102 | 0.096 | 0.153 | 0.154 | 0.155 | 0.117 | 0.103 | 0.101 |
| LLaMA-3 (8B) (lay 24) | 0.090 | 0.053 | 0.047 | 0.106 | 0.113 | 0.095 | 0.146 | 0.149 | 0.148 | 0.114 | 0.105 | 0.097 |
| LLaMA-3 (8B) (lay 30) | 0.082 | 0.066 | 0.060 | 0.120 | 0.117 | 0.101 | 0.147 | 0.151 | 0.148 | 0.117 | 0.111 | 0.103 |

## A.3 Prompts

### A.3.1 Prompts for question generation

**Prompt 1** *Generate a bulleted list of 500 diverse, non-overlapping questions that can be used to classify an input based on its semantic properties. Phrase the questions in diverse ways.*

*Here are some example questions:*
*{{examples}}*
*Return only a bulleted list of questions and nothing else*

**Prompt 2** *Generate a bulleted list of 100 diverse, non-overlapping questions that can be used to classify sentences from a first-person story. Phrase the questions in diverse ways.*

*Here are some example questions:*
*{{examples}}*
*Return only a bulleted list of questions and nothing else*

**Prompt 3** *Generate a bulleted list of 200 diverse, non-overlapping questions that can be used to classify sentences from a first-person story. Phrase the questions in diverse ways.*

*Here are some example questions:*
*{{examples}}*
*Return only a bulleted list of questions and nothing else*

**Prompt 4** *Based on what you know from the neuroscience and psychology literature, generate a bulleted list of 100 diverse, non-overlapping yes/no questions that ask about properties of a sentence that might be important for predicting brain activity.*

*Return only a bulleted list of questions and nothing else*

**Prompt 5** *# Example narrative sentences*
*{{example sentences from dataset}}*

*# Example yes/no questions*
*{{example questions already asked}}*

*Generate a bulleted list of 100 specific, non-overlapping yes/no questions that ask about aspects of the example narrative sentences that are important for classifying them. Focus on the given narrative sentences and form questions that combine shared properties from multiple sentences above. Do not repeat information in the example questions that are already given above. Instead, generate complementary questions that are not covered by the example questions. Return only a bulleted list of questions and nothing else.*

**Prompt 6** *Generate more diverse questions that may occur for a single sentence in a first-person narrative story*

See exact prompts with examples in the Github repo.

### A.3.2 Prompts for question answering

**Standard prompt** *<User>: Input text: {example}*
*Question: {question}*
*Answer with yes or no, then give an explanation.*

**Few-shot prompt** *<System>: You are a concise, helpful assistant.*
*<User>: Input text: and i just kept on laughing because it was so*
*Question: Does the input mention laughter?*
*Answer with Yes or No.*
*<Assistant>: Yes*
*<User> Input text: what a crazy day things just kept on happening*
*Question: Is the sentence related to food preparation?*
*Answer with Yes or No.*
*<Assistant>: No*
*<User> Input text: i felt like a fly on the wall just waiting for*
*Question: Does the text use a metaphor or figurative language?*
*Answer with Yes or No.*
*<Assistant>: Yes*
*<User> Input text: he takes too long in there getting the pans from*
*Question: Is there a reference to sports?*
*Answer with Yes or No.*
*Answer with Yes or No.*
*<Assistant>: No*
*<User> Input text: was silent and lovely and there was no sound except*
*Question: Is the sentence expressing confusion or uncertainty?*
*Answer with Yes or No.*
*<Assistant>: No*
*<User> Input text: {example}*
*Question: {question}*
*Answer with Yes or No.*
*<Assistant>:*

See exact prompts with examples in the Github repo.

### A.4 Information retrieval details

**Optimization details** When fitting our QA-Emb model for information retrieval, we learn a single scalar per-question that is multiplied by each embedding before computing a similarity. To learn these scalars, we minimize a two-part loss. The first loss is the negative cosine similarity between each query and its similar documents. The second loss is the cosine similarity between each query and the remaining documents. We weight the first loss as 10 times higher than the second loss and optimize using Adam [105] with a learning rate of $10^{-4}$. We run for 8 epochs, when the training loss seems to plateau.

Table A4: 54 binary classification datasets along with their underlying yes/no question and corpus statistics from a recent collection [89, 90].

| Dataset name | Dataset topic | Underlying yes/no question | Examples | Unique unigrams |
|---|---|---|---|---|
| 0-irony | sarcasm | contains irony | 590 | 3897 |
| 1-objective | unbiased | is a more objective description of what happened | 739 | 5628 |
| 2-subjective | subjective | contains subjective opinion | 757 | 5769 |
| 3-god | religious | believes in god | 164 | 1455 |
| 4-atheism | atheistic | is against religion | 172 | 1472 |
| 5-evacuate | evacuation | involves a need for people to evacuate | 2670 | 16505 |
| 6-terorrism | terrorism | describes a situation that involves terrorism | 2640 | 16608 |
| 7-crime | crime | involves crime | 2621 | 16333 |
| 8-shelter | shelter | describes a situation where people need shelter | 2620 | 16347 |
| 9-food | hunger | is related to food security | 2642 | 16276 |
| 10-infrastructure | infrastructure | is related to infrastructure | 2664 | 16548 |
| 11-regime change | regime change | describes a regime change | 2670 | 16382 |
| 12-medical | health | is related to a medical situation | 2675 | 16223 |
| 13-water | water | involves a situation where people need clean water | 2619 | 16135 |
| 14-search | rescue | involves a search/rescue situation | 2628 | 16131 |
| 15-utility | utility | expresses need for utility, energy or sanitation | 2640 | 16249 |
| 16-hillary | Hillary | is against Hillary | 224 | 1693 |
| 17-hillary | Hillary | supports hillary | 218 | 1675 |
| 18-offensive | derogatory | contains offensive content | 652 | 6109 |
| 19-offensive | toxic | insult women or immigrants | 2188 | 11839 |
| 20-pro-life | pro-life | is pro-life | 213 | 1633 |
| 21-pro-choice | abortion | supports abortion | 209 | 1593 |
| 22-physics | physics | is about physics | 10360 | 93810 |
| 23-computer science | computers | is related to computer science | 10441 | 93947 |
| 24-statistics | statistics | is about statistics | 9286 | 86874 |
| 25-math | math | is about math research | 8898 | 85118 |
| 26-grammar | ungrammatical | is ungrammatical | 834 | 2217 |
| 27-grammar | grammatical | is grammatical | 826 | 2236 |
| 28-sexis | sexist | is offensive to women | 209 | 1641 |
| 29-sexis | feminism | supports feminism | 215 | 1710 |
| 30-news | world | is about world news | 5778 | 13023 |
| 31-sports | sports news | is about sports news | 5674 | 12849 |
| 32-business | business | is related to business | 5699 | 12913 |
| 33-tech | technology | is related to technology | 5727 | 12927 |
| 34-bad | negative | contains a bad movie review | 357 | 16889 |
| 35-good | good | thinks the movie is good | 380 | 17497 |
| 36-quantity | quantity | asks for a quantity | 1901 | 5144 |
| 37-location | location | asks about a location | 1925 | 5236 |
| 38-person | person | asks about a person | 1848 | 5014 |
| 39-entity | entity | asks about an entity | 1896 | 5180 |
| 40-abbrevation | abbreviation | asks about an abbreviation | 1839 | 5045 |
| 41-defin | definition | contains a definition | 651 | 4508 |
| 42-environment | environmentalism | is against environmentalist | 124 | 1117 |
| 43-environment | environmentalism | is environmentalist | 119 | 1072 |
| 44-spam | spam | is a spam | 360 | 2470 |
| 45-fact | facts | asks for factual information | 704 | 11449 |
| 46-opinion | opinion | asks for an opinion | 719 | 11709 |
| 47-math | science | is related to math and science | 7514 | 53973 |
| 48-health | health | is related to health | 7485 | 53986 |
| 49-computer | computers | related to computer or internet | 7486 | 54256 |
| 50-sport | sports | is related to sports | 7505 | 54718 |
| 51-entertainment | entertainment | is about entertainment | 7461 | 53573 |
| 52-family | relationships | is about family and relationships | 7438 | 54680 |
| 53-politic | politics | is related to politics or government | 7410 | 53393 |

