# OpenReview forum: "Crafting Interpretable Embeddings for Language Neuroscience by Asking LLMs Questions"
_NeurIPS.cc/2024/Conference — NeurIPS 2024 poster_

### Official Review · Reviewer_s4gE · 2024-07-07

**Soundness:** 3
**Presentation:** 4
**Contribution:** 4
**Rating:** 7
**Confidence:** 4

**Summary:**

This study proposes QA-embedding for natural language text for multiple downstream tasks. Early embedding methods like bag-of-words and BM-25 cannot capture nuanced semantic feature of a sentence, and recent works primarily utilize the hidden states of large language models (LLM) as text embeddings, which lack of interpretability. Comparing to these embedding architectures, QA-embedding construct interpretable representations of texts, which can ameliorate the influence of opaqueness in scientific fields. In QA-embedding model procedures, it collects evaluation questions which is relevant to specific task firstly, which is achieved by prompting GPT-4 with corresponding information and eradicate redundancy through an Elastic. Then, QA-embedding model builds a {0,1} vectors by transforming yes/no responses to 1/0 which are obtained from asking a QA model with evaluation questions. Moreover, QA-embeddings are transformed into representations suitable for downstream tasks through parameters fitted by ridge regression. To evaluate QA-embedding, they perform experiment on fMRI, information retrieval, text clustering task and outperforms baselines. Meanwhile, they mention the limitation of QA-embedding, such as computational cost and LLM inaccuracies. To comprehend the limitation in depth, they also conduct analysis and put forward possible improvements.

**Strengths:**

1. Text embedding are widely used in both research and applications, and highly interpretable embeddings can also enhance the reliability of models.
2. QA-embedding is an intuitive method that can extract the semantic feature of a sentence.
3. The proposed method is intriguing and may inspire more studies collaborating LLMs to interpretability.

**Weaknesses:**

1. Some implementation procedures in line 149-165 should be clearly described, including PCA, inverse PCA and the sampling process, to clarify the task setup.
2. Although computational cost can be alleviated by distilling, QA model’s reliability still debilitate the interpretability of QA-embedding. According to experiment result in 5.2, merely 80% dimensions of embedding vectors are reliable, which means QA-embedding cannot consistently control the accuracy of each dimension.

**Questions:**

1. Is there an analysis on the time efficiency of QA-embedding? Although you proposed solution to decrease the computational cost, a comparison among before and after distilling and other embedding methods will make the limitation clearer.
2. In section 5.2, did you test the classification accuracy on your distilled model RoBERTa? It seems that you treat distilled RoBERTa-base as the QA model in previous experiment, and you prove distilled RoBERTa’s ability on fMRI task, thus its classification skill or behaviour on other tasks can demonstrate the effect of your distillation technique.
3. Since QA-embedding is strongly interpretable, have you tried to make error analysis on downstream tasks? The aptitude of tracing back to the essence of model’s prediction errors will make QA-embedding competitive.

**Limitations:**

Yes

---

> ### Author Rebuttal · Authors · 2024-08-07
>
> Thank you for your time and thoughtful comments - they have helped us to improve the paper.
>
> **W1. Re: implementation procedures** - thanks, we have added some notes on the task setup as well as an additional section A.3 describing the full fMRI data collection details and preprocessing.
>
> **W2. Re: computational cost** - this is a good point, the QA model’s interpretability relies on (i) the underlying LLM being sufficiently capable and (ii) the questions being asked to construct the embedding not being too difficult. The experiments in Sec 5.2 show that different LLMs get ~80% on a set of datasets that are both difficult and noisy. We anecdotally have two of our paper’s authors answer 100 questions at random on this dataset and they achieve 84% and 78%. We describe one of the difficulties in the dataset in Sec 5.2:
>
> “the dataset yielding the lowest accuracy asks the question Is the input about math research?. While this may seem like a fairly simple question for an LLM to answer, the examples in the negative class consist of texts from other quantitative fields (e.g. chemistry) that usually contain numbers, math notation, and statistical analysis. Thus the LLMs answer yes to most examples and achieve accuracy near chance (50%).”
>
> For the main fMRI setting we focus on, these difficulties are rare & answering these questions is much simpler, especially since the fMRI input lengths are each 10 words, whereas the input lengths for the datasets in Sec 5.2 are over 50 words on average (with some inputs spanning over 1,000 words).
>
> **Q1. Re: time efficiency** – Thanks for pointing this out, we have added some discussion on time efficiency to Section 2 (the methods). Generally, if an embedding requires answering *n* questions, the cost of the full autoregressive approach is *n* times the inference cost of the large autoregressive model. The cost of the distilled approach is the inference cost of the smaller distilled model (plus the negligible cost of a single linear layer for classifying the answer to each question), thus reducing the total cost by more than a factor of *n*.
>
> **Q2. Re: distilled model classification accuracy** – We did not directly test the classification accuracy of the distilled RoBERTa model (although many prior studies have shown that RoBERTa can be distilled to mimic larger teacher models). In a new experiment, we measure the distilled model’s ability to mimic the original LLaMA-based full model. Specifically, we directly replace the embedding obtained from the full model with the embedding obtained from the distilled model *without changing the linear weights learned on top of the embeddings*. We find that fMRI prediction performance drops from a mean test correlation of 0.114 to 0.112. This very small drop suggests that the distilled embeddings are a drop-in replacement for the original interpretable embeddings that preserve the correct, interpretable values for different questions in the embedding.
>
> **Re: Flag for Ethics review** – the fMRI dataset we study is publicly available and comes from two recent studies (LeBel et al., 2022; Tang et al., 2023). In both cases, the experimental protocol was approved by the Institutional Review Board at the University of Texas at Austin and written informed consent was obtained from all subjects.

---

### Official Review · Reviewer_5mep · 2024-07-10

**Soundness:** 2
**Presentation:** 3
**Contribution:** 3
**Rating:** 5
**Confidence:** 3

**Summary:**

The paper explores obtaining interpretable embeddings through LLM prompting. To address the opaque nature of text embeddings, the authors introduce question-answering embeddings (QA-Emb) by asking LLMs a set of yes/no questions. Specifically, these questions are generated by GPT-4 using predefined prompts. Then, Elastic net is adopted for feature selection. The validity of QA-Emb is tested primarily in a fMRI interpretation problem. Moreover, the paper explores the computational efficiency of the method by finetuning a RoBERTa model and extends the QA-Emb on information retrieval and text clustering.

**Strengths:**

- Generating interpretable embeddings using LLMs is an important and promising issue.
- QA-Emb can potentially provide new insights, demonstrated by the interesting findings in mapping the questions' sensitivity across the cortex.
- QA-Emb shows promising results in the fMRI interpretation problem.

**Weaknesses:**

- The generation of questions for QA-Emb relies on manually crafted prompts, raising concerns about the robustness of the method. It remains unclear whether the efficacy of QA-Emb would persist with different sets of prompts. Moreover, The inconsistency in the number of questions generated from each prompt (500, 100, 200, 100, 100 respectively) introduces additional complexity. It remains unclear how to determine the optimal number of generated questions for each prompt.
- The method is computationally intensive, as it requires prompting LLMs with hundreds of questions for each sample. While the authors attempt to mitigate this by fine-tuning RoBERTa, it is questionable whether a smaller model like RoBERTa can maintain performance across diverse applications with more complex questions. Furthermore, the fine-tuning process itself depends on prompting LLMs over substantial examples.
- The demonstrated effectiveness of QA-Emb is limited to a single neuroscience problem. Its performance in information retrieval and text clustering appears underwhelming. The approach would gain more credibility if shown effective across more scenarios.

**Questions:**

Could you provide more details on the method and prompt for the posthoc pruning of generated questions in the experiments?

**Limitations:**

Limitations are included.

---

> ### Author Rebuttal · Authors · 2024-08-07
>
> Thank you for your time and thoughtful comments - they have helped us to improve the paper.
>
> **(1) Re: prompt sensitivity** – Indeed prompting LLMs for this application requires some manual choices, although this can be a good thing for helping to inject domain knowledge into the problem (different numbers of desired questions can themselves be put into the prompts). We conduct new experiments extending the results in Table A2 to assess prompt sensitivity and find that results are fairly stable to the choices of different prompts. We report the mean fMRI test correlation, averaged across the test set for all three subjects. See the exact prompts in Sec A.3.1.
>
> We find that performance does not vary drastically based on the choice of prompt:
>
> |Prompts    |1    |2    |3    |4    |5   |6   |
> |-----------|-----|-----|-----|-----|----|----|
> |Performance|0.088|0.092|0.085|0.079|0.90|0.96|
> |Questions  |90   |98   |88   |97   |45  |156 |
>
> We perform feature selection as done in the main paper (running multi-task Elastic net with 20 logarithmically spaced regularization parameters ranging from $10^{−3}$ to 1 and then fitting a Ridge regression to the selected features) and report results for the model with number of features closest to 29 (which is the main model we analyze in the paper). We find that performance again does not vary drastically based on the choice of prompt:
>
> |Prompts    |1    |2    |3    |4    |5   |6   |
> |-----------|-----|-----|-----|-----|----|----|
> |Performance|0.062|0.065|0.059|0.061|0.072|0.074|
> |Questions  |20   |24   |17   |33   |26  |34  |
>
> Finally, we find that performance again does not vary drastically as the number of questions becomes large. Table A2 provides further breakdowns for this table by underlying LLM
>
> |Prompts    |1-3  |1-5  |1-6  |
> |-----------|-----|-----|-----|
> |Performance|0.111|0.115|0.116|
> |Questions  |376  |518  |674  |
>
>
> **(2) Re: computational cost** – The proposed method is computationally intensive, although for the main application we study here (the fMRI results), distilling RoBERTa successfully cut these costs and should similarly work in situations where questions are easy to answer. In the long run, we hope costs will be mitigated by (1) the rapidly decreasing costs of LLM inference and (2) the improvement of small models which can be used for distillation.
>
> **(3) Re: neuroscience focus** – We agree this result is specific to the neuroscience problem and have revised the manuscript and generality claims to focus on the fMRI setting (see our “General response” comment); this approach is a strong fit for the fMRI problem and reveals new insights into language neuroscience. We hope that NeurIPS reviewers for a paper in the “Neuroscience and cognitive science” track can appreciate the paper’s results.

---

> > ### Comment · Reviewer_5mep · 2024-08-13
> > **Response to the Rebuttal**
> >
> > I appreciate the authors’ responses during the rebuttal period. However, most of my concerns remain unresolved. The performance appears to vary in a wide range based on the choice of prompt, and the computational cost is still a concern. Additionally, I am skeptical about the decision to shift the focus of the paper towards neuroscience. While this shift may mitigate QA-Emb’s performance limitations in other tasks, it also diminishes the contribution of this work in generating interpretable embeddings for a broader context.
> >
> > Overall, this paper tackles the important issue of generating interpretable embeddings by LLMs, but it also presents clear weaknesses. Therefore, I decide to maintain my original assessment, which reflects my intention to encourage the exploration in this underexplored direction.

---

### Official Review · Reviewer_Z19r · 2024-07-12

**Soundness:** 2
**Presentation:** 3
**Contribution:** 2
**Rating:** 3
**Confidence:** 4

**Summary:**

The paper proposes a method of prompting LLMs with a list of yes or no questions to obtain binary embeddings for texts. The list of questions is generated by prompting the LLM with task-specific knowledge. The proposed method, QA-Emb, is evaluated primarily on predicting fMRI voxel responses to texts. QA-Emb outperforms an established interpretable baseline but underperforms compared to a black-box LLM embedding baseline. Additional experiments in information retrieval indicate that QA-Emb significantly underperforms BM25 but performs better than simple n-grams.

**Strengths:**

1. The paper introduces a creative perspective on text embedding by leveraging the QA capabilities of LLMs to achieve interpretability.
2. The paper is well-written and easy to follow. It includes extensive experiments demonstrating the effectiveness and limitations of the method in various scenarios.
3. This paper represents a initial effort in crafting an interpretable embedding space for tasks like regression, information retrieval, and clustering.

**Weaknesses:**

1. The method of prompting LLMs to generate interpretable features is not entirely novel. For example, "CHiLL: Zero-shot Custom Interpretable Feature Extraction from Clinical Notes with Large Language Models" also prompts LLMs to ask questions for a classification task. A deeper investigation into the question generation phase would be more interesting and challenging. The paper could benefit from a more detailed discussion on systematically selecting and validating these questions to ensure robustness across various tasks.

2. While the method performs well in predicting fMRI voxel responses, it significantly underperforms in the information retrieval task compared to BM25. This raises doubts about the method's generalizability to other domains and tasks. If the method only excels in predicting fMRI voxel responses, it may be more accurately described as a specialized model for this task rather than a general "interpretable embedding" model.

3. The QA-Emb pipeline is computationally intensive, requiring multiple LLM calls to compute an embedding, which can be prohibitively expensive.

**Questions:**

1. More experiments on different domains and tasks could help to show the generalizability of the method. Especially, on embedding related benchmarks, and against some interpretable and black-box baselines.

2. What strategies can be employed to mitigate the computational cost of QA-Emb?

**Limitations:**

Yes.

---

> ### Author Rebuttal · Authors · 2024-08-07
>
> Thank you for your time and thoughtful comments - they have helped us to improve the paper.
>
> **1) Re: question generation** – We wholeheartedly agree question generation is important and have added new experiments extending the results in Table A2 to analyze the process of question generation. We find that results are fairly stable to the choices of different prompts. We report the mean fMRI test correlation, averaged across the test set for all three subjects. See the exact prompts in Sec A.3.1.
>
> We find that performance does not vary drastically based on the choice of prompt:
>
> |Prompts    |1    |2    |3    |4    |5   |6   |
> |-----------|-----|-----|-----|-----|----|----|
> |Performance|0.088|0.092|0.085|0.079|0.90|0.96|
> |Questions  |90   |98   |88   |97   |45  |156 |
>
> We perform feature selection as done in the main paper (running multi-task Elastic net with 20 logarithmically spaced regularization parameters ranging from $10^{−3}$ to 1 and then fitting a Ridge regression to the selected features) and report results for the model with number of features closest to 29 (which is the main model we analyze in the paper). We find that performance again does not vary drastically based on the choice of prompt:
>
> |Prompts    |1    |2    |3    |4    |5   |6   |
> |-----------|-----|-----|-----|-----|----|----|
> |Performance|0.062|0.065|0.059|0.061|0.072|0.074|
> |Questions  |20   |24   |17   |33   |26  |34  |
>
> Finally, we find that performance does not vary drastically as the number of questions becomes large. Table A2 provides further breakdowns for this table by underlying LLM.
>
> |Prompts    |1-3  |1-5  |1-6  |
> |-----------|-----|-----|-----|
> |Performance|0.111|0.115|0.116|
> |Questions  |376  |518  |674  |
>
> While the method of “prompting LLMs to generate interpretable features” may not be entirely novel, we believe our method and application are. Unlike the ChiLL paper, the LLM itself (i) writes the questions and (ii) a feature selection process prunes them. This allows for the method to discover a small set of underlying important features, rather than a human specifying them. This is critical in the fMRI setting, where specifying a small, interpretable encoding model has long been pursued without success. In contrast, QA-Emb can yield very strong performance with only 29 questions (that would have been difficult for the domain experts to explicitly select in the first place). We have added a discussion of the ChiLL paper and clarified that QA-Emb’s novelty lies in (i), (ii), and the new insights they reveal in the fMRI setting.
>
>
> **2) Re: embedding generality** - Thanks, we take this point well and have revised the manuscript and generality claims to focus on the fMRI setting (see our “General response” comment). This fMRI problem is the main motivation for us, where interpretability is crucial and a minor performance drop is acceptable. We hope that NeurIPS reviewers for a paper in the “Neuroscience and cognitive science” track can appreciate the paper’s results. In settings outside of fMRI, the method does underperform baselines, although it still succeeds in improving the baseline through concatenation (and also provides very small embeddings, which may be useful in niche settings).
>
> **3) Re: computational efficiency** - The pipeline is indeed computationally expensive, but we propose and evaluate a solution: model distillation. The intro/methods briefly describe it, e.g. “we find that we can drastically reduce the computational cost of QA-Emb by distilling it into a model that computes the answers to all selected questions in a single feedforward pass by using many classification heads.” This makes computing an embedding inexpensive at inference time (see Sec 5.1) with a negligible drop in performance in our fMRI setting (see Table 1, where mean test correlation drops from 0.114 to 0.113 after distillation).
>
> Distillation also does not substantially hurt the interpretability of the underlying embedding elements. In a new experiment, we measure the distilled model’s ability to mimic the original LLaMA-based full model: we directly replace the embedding obtained from the full model with the embedding obtained from the distilled model *without changing the linear weights learned on top of the embeddings* for the fMRI setting. We find that performance drops from a mean test correlation of 0.114 to 0.112. This extremely small drop suggests that the distilled embeddings are a drop-in replacement for the original interpretable embeddings that preserve the correct, interpretable values for different questions in the embedding.

---

> > ### Comment · Reviewer_Z19r · 2024-08-13
> >
> > Thanks for your response.

---

### Official Review · Reviewer_wvR8 · 2024-07-19

**Soundness:** 2
**Presentation:** 3
**Contribution:** 2
**Rating:** 5
**Confidence:** 3

**Summary:**

The authors introduce question-answering embeddings (QA-Emb), where each feature in the embedding corresponds to an answer to a yes/no question asked to an LLM (e.g., LLaMA-3 8B). QA-Emb significantly outperforms an established interpretable baseline in predicting fMRI voxel responses to language stimuli. The authors also evaluate QA-Emb on information retrieval and text clustering tasks. Additionally, computational efficiency can be improved by distilling the QA-Emb model into a RoBERTa-base model.

**Strengths:**

(1) This paper focuses on the critical problem of extracting interpretable embeddings from large language models (LLMs).

(2) Extensive experiments are conducted on three different tasks: fMRI interpretation, information retrieval, and text clustering.

(3) The authors demonstrate that the model's efficiency can be improved through distillation.

**Weaknesses:**

(1) Motivation:
The paper aims to extract meaningful text embeddings from LLMs. However, the entire QA-Emb pipeline seems to rely less on LLMs than suggested, as these questions could also be answered by a much smaller QA model, such as RoBERTa-base, which the authors have explored in Section 5.1.

(2) Baselines:
While there are existing methods that extract text embeddings from LLMs [1,2], these methods are not compared in the paper. It is reasonable to expect that QA-Emb might perform worse than these non-interpretable methods, but a comparison with these methods would provide a clearer understanding of QA-Emb's performance.

(3) Results:
The authors claim that QA-Emb is a high-performing embedding method. However, results on the MS-MARCO tasks indicate otherwise. QA-Emb significantly lags behind BM25, while a recent LLM-based embedding method like PromptReps [1] is able to match the performance of BM25 on retrieval tasks.

[1] Ting Jiang, Shaohan Huang, Zhongzhi Luan, Deqing Wang, Fuzhen Zhuang. Scaling Sentence Embeddings with Large Language Models. Arxiv.

[2] Shengyao Zhuang, Xueguang Ma, Bevan Koopman, Jimmy Lin, Guido Zuccon. PromptReps: Prompting Large Language Models to Generate Dense and Sparse Representations for Zero-Shot Document Retrieval. Arxiv.

**Questions:**

Please see my comments in Weakness.

**Limitations:**

The authors provide detailed discussions on limitations in Sections 2 and 5.

---

> ### Author Rebuttal · Authors · 2024-08-07
>
> Thank you for your time and thoughtful comments - they have helped us to improve the paper.
>
> (1) Re: Motivation – Yes indeed, the entire pipeline at inference-time can use only smaller models (e.g. RoBERTA). We see this is a major strength, as it drastically reduces the inference cost of applying QA-Emb. However, at training time, a high-performing LLM (e.g. LLaMA-3) is still required to provide the answers that can be used to finetune the smaller model.
>
> (2) Re: Baselines – Thanks for sharing these baselines, we have added references to them in the paper. Indeed there are many non-interpretable baselines that we can compare to. For our main setting (fMRI prediction), state-of-the-art prediction empirically comes from embeddings obtained by taking the last token of state-of-the-art open source models such as LLaMA [1]. We do compare directly to these across many layers of recent LLaMA models (Fig 2A). We will add these comparisons for the information retrieval setting in Table 2 as well (we do expect the non-interpretable embeddings to perform better in this setting).
>
> (3) Re: Results – Thanks for pointing this out, indeed QA-Emb is a strong embedding method in terms of interpretability/compactness of the representation but yields a small drop in performance compared to black-box methods. This tradeoff is worthwhile in our fMRI setting, where the goal is to yield a succinct model for scientific understanding. In our main result (Fig 2B), QA-Emb yields very strong fMRI prediction performance using only a 29-dimensional binary representation that then enables manual inspection for scientific understanding (Fig 3). Note that we have re-worked the paper to be focused on the fMRI setting (see “General response” comment). We hope that NeurIPS reviewers for a paper in the “Neuroscience and cognitive science” track can appreciate the paper’s results.
>
> Representations from baseline methods are often black-box or very large. PromptReps embeddings do yield high performance, but contain an element for each token in an LLM’s vocabulary, yielding a representation with more than 100k dimensions in the PromptReps paper’s main results (with LLaMA-3).
>
> [1] “Scaling laws for language encoding models in fMRI” R Antonello, A Vaidya, A Huth, NeurIPS 2023

---

### Author Response · Authors · 2024-08-07
**General response**

We thank the reviewers for their thoughtful comments. We have made several changes to the paper in the revised version to incorporate the feedback, resulting in a much improved paper. The biggest change is to better focus the paper on the fMRI results, which are the main experiments and motivation for the paper. In the fMRI literature, interpretable encoding models are extremely desirable, but have largely been displaced by LLM-based black-box models that yield superior predictive performance. The QA-Emb approach is an effective way of yielding interpretable models that roughly match SOTA fMRI prediction performance and is a wholly novel approach in the encoding model space .

Our changes include edits throughout the paper and a title change to “Crafting Interpretable Embeddings **for Language Neuroscience** by Asking LLMs Questions”. These changes are relatively minor framing changes, as the results / experiments throughout are already focused on this application (everything but Sec 6). For example, the introduction reads:

> Grounding in a neuroscience context allows us to avoid common pitfalls in evaluating interpretation methods [19, 20] that seek to test “interpretability” generally. Additionally, this focus allows to more realistically integrate domain knowledge to select and evaluate the questions needed for QA-Emb, one of its core strengths.

The reviewers seem to agree that the proposed method is a useful contribution to the area of fMRI language neuroscience. We hope that in this case, the paper is suitable for publication at NeurIPS, as NeurIPS remains a major venue for neuroscience work, and this particular paper is a submission to the “Primary Area: Neuroscience and cognitive science” with Keywords “fMRI, Encoding models, Neuroscience, Language neuroscience, Interpretability, Large language models, Explainability, Brain mapping”.

---

### Decision · Program_Chairs · 2024-09-25

**Decision:**

Accept (poster)

**Comment:**

The paper proposes an approach QA-Emb for generating interpretable embeddings (with the primary application being fMRI response prediction where interpretability is highly valued).


They create this embedding based on prompting an LLM with a set of yes/no questions. This yes/no question strategy has been used before in fMRI but the authors' contribution is that they use an LLM to generate these questions and also automatically prune them (so they do not have to be manually constructed).

I think this is a nice application of LLM prompting the fMRI and so support accepting the paper contigent on the authors' proposed title change. (The NLP results of the paper are not strong but I think the authors intended the main application to be about fMRI).